# Mechanism of and Key Technologies for Copper Bonding in the Hot Rolling of SCR Continuous Casting and Rolling

**Yang Liu** [1] , **Yan Peng** [1,*] **and Xiaobo Qu** [2]

1   National Engineering Research Center for Equipment and Technology of Cold Strip Rolling, Yanshan University, Qinhuangdao 066004, China; liuyang863@aliyun.com
2   Jiangsu Yonggang Group Co. Ltd., Suzhou 215628, China; qubo6101@163.com
*   Correspondence: pengyan@ysu.edu.cn

**Abstract:** In the process of copper alloy hot continuous rolling, the problem of copper sticking to the roller seriously affects the surface quality, performance, and service life of the copper products. Roll sticking occurs as the adhesion energy of Cu is lower than that of Fe and the Fe-Cu interface, and the severe surface deformation which forces the copper into direct contact with the roll during the process of profile rolling. Based on the copper deformation law and adhesion phenomenon in the hot continuous rolling process, a rolling deformation model and roll copper adhesion model or copper alloy hot continuous rolling were established, and their simulation was realized using finite element software. Through finite element modeling of the hot rolling deformation zone, the distribution of the temperature, contact normal stress, and exposure rate in the hot rolling deformation zone were obtained, which were consistent with the actual roll adhesion phenomenon and copper adhesion position. To address the copper sticking behavior of the rolls, the process optimization method of matching the motor speed with the elongation coefficient (the 1# and 2# motor speeds were adjusted to 1549 r/min and 1586 r/min, respectively), adjusting the roll gap to 7.9 mm, and increasing the number and pressure of roll spray nozzles were put forward, which effectively solved the problem of copper sticking to the roll, significantly improved the surface quality of the copper and the service life of the roll, and can be used in production.

**Keywords:** copper rod; sticking behavior; hot continuous rolling; rolling process parameters

## 1. Introduction

Copper and its alloys are important non-ferrous metals. Due to their excellent overall performance (higher corrosion resistance, excellent electrical and thermal conductivity, easy manufacturing, resistance to biological pollution, and other attractive features), they are widely used in many environments and industries [1]. Over recent years, with the rapid development of the Chinese economy and industrial technology, the development of advanced material connection technology is the key to expanding the potential applications of engineering alloys, with copper alloy in particular exhibiting significant potential. The Ni-Ti-Cu alloy induces the formation of a nano-level transition layer due to the dynamic recrystallization and high strain rate of the Cu foil, which increases the metallurgical bonding performance and enhances the strength [2]. Copper-based shape memory alloys (Cu-Al-Mn) have potential applications in earthquakes, due to their reduced cost, excellent superelasticity, and improved strength and energy absorption [3]. At the same time, due to better control of the microstructure, copper-based shape memory alloys have lower production costs and higher mechanical properties, and so are a potential substitute for Ni-Ti shape memory alloys [4]. As the main raw material of wire and cable, and the main transportation carrier of various copper alloys, copper wire is widely used in the fields of power, electronics, machinery construction, and more. It is regarded as the "blood vessel" and "nerve" of the national economy [5]. In the process of hot continuous rolling of copper wire, the problem of copper sticking to rolls has an important impact on the final product

specifications, and on the rolling process itself. The surface adhesion phenomenon of the roller is such that the surface damage shortens the service life of the roller, resulting in poor surface quality of the copper rod, resulting in an unnecessary waste of energy and low production efficiency. Therefore, the problem of copper sticking on rolls has become a key constraint in the production of high-quality copper wire.

The problem of rolled products sticking to the surface of rolls exists in the rolling of both ferrous and non-ferrous metals. The present study aiming to address the problem of copper sticking to the roll during the hot rolling process, by investigating the key technologies intended to improve the production line process of high quality and large output continuous casting and rolling of copper rod. Alongside wealth of knowledge concerning continuous casting and rolling technology and production experience in iron and steel, there also exists significant research on the mechanism of sticking rollers. Xiaomi Cheng [6] studied the influence of different surface characteristics (smooth surface, surface with 45° grinding mark, and sand surface with oscillation marks) on the oxidation and tribological properties during hot rolling, which indicated that the surface morphology had an important influence on the integrity of oxide film during hot rolling. Sun Bin [7] analyzed the structure and formation mechanism of adhesive spot defects on the surface of medium and heavy plates. Through the analysis of the formation of the adhesive defects and the detection of the adhesive elements, he proposed an effective method to control the quality of the roll surface and eliminate the adhesive spot defects. Yang Lianhong [8] analyzed the mechanism underlying bonding behavior during hot rolling, discussed the hot rolling process conditions affecting the bonding behavior, and proposed the mechanism of bonding nucleation and growth. S I Platov et al. [9] studied the operation of the roll system when a liquid lubricating material was applied during the hot rolling process, and demonstrated that the lubricating material reduced the wear of the work roller by 10–12%, effectively reducing the sticking phenomenon. Zhang Chi et al. [10] discussed the behavior of sticking hot rolling deformation rollers, and suggested that sticking rollers occur due to a process of nucleation, growth, and saturation, and that micro-cracks are needed to provide the necessary nucleation position for adhesion of the roller surface. Peng Yan et al. [11] utilized a process simulation and quality control of copper rod continuous casting and rolling, and established the finite element simulation of copper rod continuous casting and hot rolling process in the SCR3000 continuous casting and rolling line. Chol J.Y [12] studied the nucleation and growth process of bonded particles in ferritic stainless steel, which revealed that the behavior of roller sticking was closely related to the surface roughness and oxidation resistance of the roller. Guo Hesong et al. [8] proposed to analyze the relationship between reduction rate and rolling pressure under different lubrication conditions by using the lubrication principle of hot rolling. Through the theoretical calculation of rolling interface lubrication, the spray device has been improved to solve the failure problem of the 10# roller, and the surface quality of the copper rod has also been raised. Guo et al. [13] proposed an optimized lubrication process and spray device by calculating the lubrication performance of the copper rod hot tandem rolling roll to improve the roll performance, the roll joining performance, and the quality of the copper rod. C Jin-Won et al. [14] have carried out a detailed study on steel-bonding behavior under hot rolling conditions, which revealed that the high-temperature tensile strength and oxidation resistance are important factors affecting bonding between the bare metal and the roll surface.

The SCR production line is favored by copper rod manufacturers around the world, due to its stable product quality, high performance, good efficiency, equipment structure, simple operation, and easy maintenance [15]. The continuous casting and rolling process directly determine the working capacity of the SCR copper bar production equipment, as well as the quality and performance of the finished copper bars. By simulating and analyzing the microstructure and composition of copper alloy produced by the SCR method, Yang [16] theoretically established the relationship between microstructure and macro properties, and fully revealed the solidification behavior and property changes of copper alloy. When changing the numerical simulation parameters, Sahoo [17] studied the effects

of casting speed and temperature on the solidification process of continuous casting by considering the selection of solidification and turbulence models. Komanduri et al. [18] studied the mechanical properties of copper, such as tensile strength, and explained the influence of deformation speed and degree on the deformation process of fine-grained copper rods. In addition, the adhesion performance of rolled products in the rolling deformation zone was found to be comprehensively affected by the rolling material, reduction rate, roll speed, sliding friction factor, roll surface roughness, and lubrication performance. During hot rolling, the adhesion mechanism is affected by high-temperature mechanical and oxidation properties, and the rolling temperature has a great impact on the bonding of rolled metal [19]. At the same time, adhesion can be minimized by increasing lubrication, rolling speed, and rolling temperature [20,21]. Despite significant research into the adhesion properties and mechanism of the rolling process, a model corresponding to the adhesion mechanism of rolling deformation of non-ferrous metals, such as copper, has yet to be elucidated. Therefore, the sticking behavior and mechanism underlying the hot continuous rolling of copper wire, which have important practical significance in solving the problem of copper sticking to the hot rolls, require further research to promote the production and manufacture of high-quality copper wire. In this paper, the hot rolling deformation behavior of the SCR production line copper wire rod and the copper sticking phenomenon of the 2# roll are numerically analyzed. According to the mechanism of copper sticking, the failure position of shear fracture is determined, and, the adhesion index model and the proportion model of bonding area are established and solved to reveal the physical nature and mechanical behavior of the copper sticking to the roll. Additionally, a simulation analysis of the rolling state and the rolling interface state of the 2# continuous rolling process expounds the influence of the hot rolling process conditions of the copper wire on copper sticking to the roll. By optimizing the existing process, positive results were achieved which may result in economic benefits in this enterprise.

## 2. Hot Rolling Process of Copper Alloy

South Wire Company developed the first SCR method for copper rod continuous casting and rolling production line. In the rolling process production line, the Morgan double-roll stand is adopted, the rough rolling stand employs the box-circle type system, and the finishing stand adopts the "elliptical circle" system [22]. Ten Morgan two-high cantilevered tandem mills are used in the hot rolling of the SCR3000 production line, which are arranged according to the level interchange. A schematic diagram of the hot continuous rolling process is shown in Figure 1. The actual realization of deformation and various process parameters during the rolling process of the specific 8 mm copper wire rod are presented in Table 1. The rolling pieces pass through each mill at the same time and are rolled according to the ellipsoidal hole type system. Each frame is driven separately. During the continuous rolling process, the rolling pieces realize a continuous and torsion-free state, and the rolling pieces automatically bite in, greatly improving production efficiency. Copper rods for electricians are mainly T2 copper (copper-silver alloy) [23]. The specific chemical composition is listed in Table 2.

The Cu–Ag alloy microstructure under different rolling passes is shown in Figure 2. The average grain size of the first pass is about 120 $\mu$m, and after 10 passes, it is approximately 20$\mu$. (d-g) illustrate the typical changes in the microstructure of the Cu–Ag alloy in detail [24].

The continuous casting billet is a copper billet with a trapezoidal cross-section and an area of 3800 mm2. The billet temperature is 960 °C. The billet is cold and edge-cut (by at most $3 \times 45°$) at two angles of continuously cast billets (at the major base of the trapezoid). The continuous casting billet enters the rolling process at 870 °C and 0.2110 m/s. The purpose of the cast slab trimming is to reduce the number of defects that may cause the wire to appear to have "flying fins" [25].

In the initial stage of intermediate roll change in the SCR production line, there was no problem with copper sticking to the roll. After a period of equipment operation, the

phenomenon of copper sticking to the two rolls of the rolling mill began to appear, and became more and more serious over time, as shown in Figure 3.

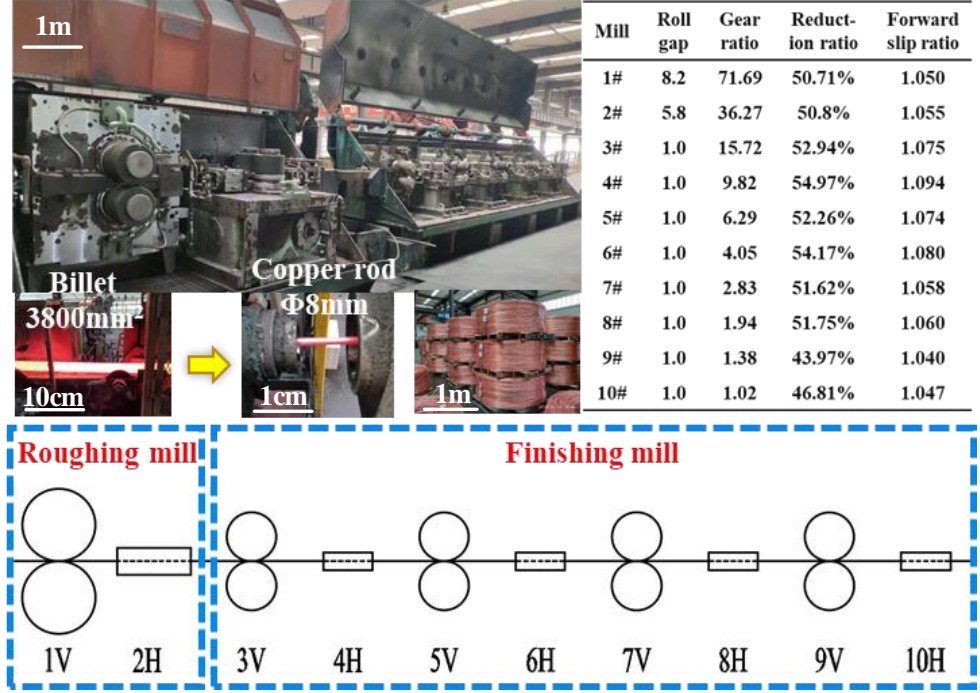

**Figure 1.** Schematic diagram of copper rod hot continuous rolling.

**Table 1.** Actual Realized Conditions of Deformation in the Process of Rolling of Copper Wire Rods of 8 mm Diameter.

| Group of Stands | No. of the Pass | Shape of the Billet/Pass | Bar Wire Size | | | Absolute Reduction, mm | Absolute Spread, mm | Relative Reduction,% | Wide Spread | Friction Coefficient |
| | | | Height, mm | Width, mm | Cross-Sectional mm² | | | | | |
|---|---|---|---|---|---|---|---|---|---|---|
| Billet | | Trapezoid | 51 | 72.45 | 3800 | – | – | – | – | – |
| Rough | 1V | Plane oval | 32.2 | 88 | 2500 | 19.8 | 15.5 | 50.71 | 1.215 | 0.645 |
| | 2H | Circle | 43.3 | 45.9 | 1320 | 44.7 | 14.7 | 50.8 | 1.471 | 0.655 |
| Finishing | 3V | Oval | 21.6 | 57.3 | 800 | 24.3 | 14 | 52.94 | 1.323 | 0.665 |
| | 4H | Circle | 25.8 | 27.23 | 480 | 31.5 | 5.63 | 54.97 | 1.261 | 0.68 |
| | 5V | Oval | 13 | 35.57 | 300 | 14.23 | 9.77 | 52.26 | 1.379 | 0.69 |
| | 6H | Circle | 16.3 | 16.95 | 195 | 19.27 | 3.95 | 54.17 | 1.304 | 0.7 |
| | 7V | Oval | 8.2 | 24.25 | 125 | 8.75 | 7.98 | 51.62 | 1.488 | 0.705 |
| | 8H | Circle | 11.7 | 11.78 | 95 | 12.55 | 3.58 | 51.75 | 1.437 | 0.715 |
| | 9V | Oval | 6.6 | 15.04 | 65 | 5.18 | 3.34 | 43.97 | 1.285 | 0.725 |
| | 10H | Circle | 8 | 8.28 | 50.27 | 7.04 | 1.68 | 46.81 | 1.255 | 0.735 |

**Table 2.** Chemical composition of T2 copper (mass fraction, %).

| Element | Cu | Fe | Pb | S | O | Sn | As | Sb | Ni | Bi | Zn |
|---|---|---|---|---|---|---|---|---|---|---|---|
| Mass fraction, % | 99.9 | 0.005 | 0.005 | 0.005 | 0.06 | 0.002 | 0.002 | 0.002 | 0.002 | 0.002 | 0.005 |

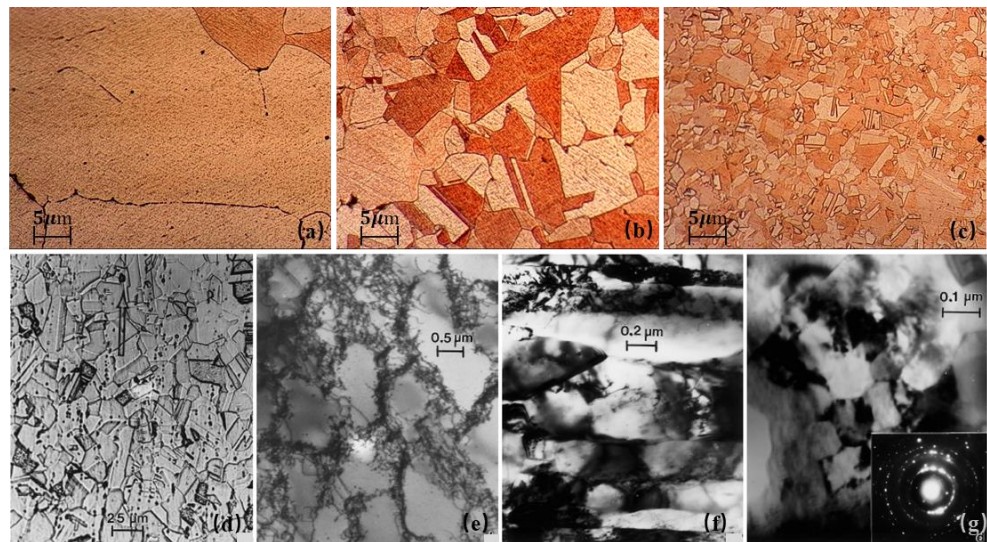

**Figure 2.** Surface morphology of T2 copper in different rolling passes: (**a**) billet; (**b**) 1 pass; (**c**) 10 pass; (**d**) light microscope view along a longitudinal half-section of a commercial copper rod; (**e**) TEM bright-field image of dislocation cells corresponding to the longitudinal view in a; (**f**) TEM bright-field image of longitudinal section; (**g**) TEM bright-field image for transverse section corresponding to c.

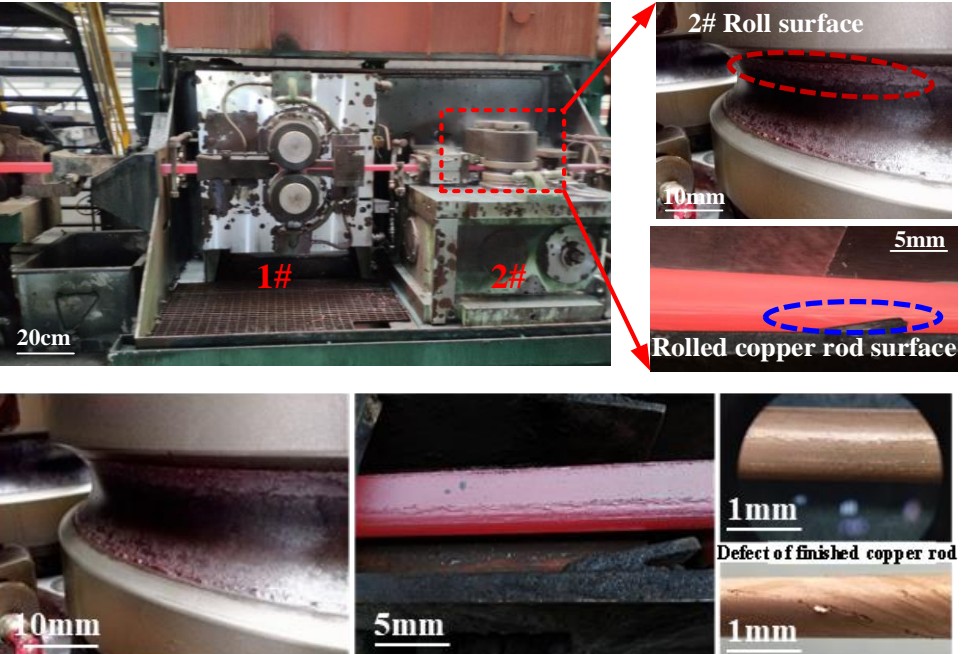

**Figure 3.** Schematic diagram of copper sticking to the 2# stand roll and wear of the rolled copper rod.

It can be seen from Figure 3 that copper sticking occurs on the upper and lower contact surfaces of the roller. There is less adhesion on the inner side, less adhesion on the outer edge, and more adhesion on the middle part of the sticking area. Pits can be seen on the edge of the copper rod (see Figure 4) rolled by the sticking roller. In the process of producing copper rods for electrical purposes, once the problem of copper sticking on the roller occurs, the service life of the roller is significantly shortened, and the surface quality of the copper rod product is also affected, as the amount of copper powder and the wire breakage rate both increase in the subsequent wire drawing process. An accumulation of copper in the groove of the roller forms bumps on the surface, which deteriorates the surface quality of the finished copper wire. Therefore, the amount of copper stuck to the

surface of the roll is the main criterion to determine whether the finishing groove needs to be replaced.

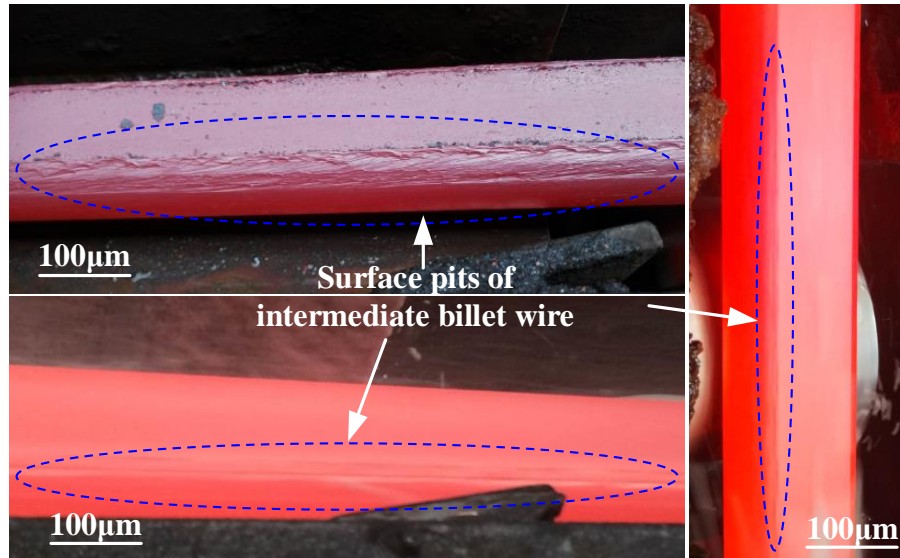

**Figure 4.** Schematic diagram of pits on the surface of the copper wire rod of the intermediate billet.

## 3. Theoretical Modeling and Research

During the hot rolling process, the high temperature on the surface of the roll promotes the formation of micro-cracks on the surface of the rolled material. When oxygen diffuses to the micro-crack area, their outer surface is strongly oxidized. At the same time, the effects of variable thermal and mechanical stress on the surface of the roll cause rapid destruction and peeling of the oxide on the surface of the rolled material, exposing its substrate [26]. The problem of roll surface adhesion during rolling is due to the relative sliding between the roll and the rolling stock in the contact area. Under high rolling pressure, the rough roll surface will scratch the surface of the rolled piece, resulting in the phenomenon of roll surface adhesion. Sticking of the roller itself is based on the gradual loss and migration of contact surface materials, accompanied by the existence of sliding friction [27]. In this paper, roller sticking is considered to be a tribological phenomenon, as research concerning the mechanism of roller sticking is also based on tribological principles. Meanwhile, the deformation and adhesion friction of a hot-rolled copper rod are analyzed in detail, and the behavior of copper sticking to the roller is revealed by establishing models of each sticking roller.

### 3.1. Mechanism of Roller Sticking to Rod in the Hot Rolling of Copper Rod

In the process of hot rolling, when the reduction ratio is large, the plastic deformation of the surface layer of the copper rod causes the brittle oxide coating on the surface of the copper rod to fracture, and the copper rod is exposed directly without protection of the oxide layer. With an increase in rolling force, the contact point between the two metals experiences local deformation, resulting in the fracture of the brittle coating, an increase in the contact surface expansion rate, cracking of the coating and extrusion of the processing material [28]. At the micro-level, the sharp plastic deformation in the rolling zone leads to the copper bar matrix extruded oxide film cracking and forming a reaction film with the polar molecules of emulsion. When the reaction film is damaged under pressure, the extruded copper matrix directly contacts the roller surface, which are welded to each other by friction, as shown in Figure 5. To reduce the surface energy of two metals in close contact, the interface bond is formed by the combination of the surface lattice. With the sheer movement of the near-contact surface, the interface bond is strengthened continuously, resulting in the phenomenon of welding [29]. As the shear load continues to

act, the interface energy formed by the interface bond increases continuously. When the critical interface energy is exceeded, the two metals undergoing welding are destroyed. The size of the interface energy determines the location of the fracture point of the contact welding. As such, the macroscopic shear strength of the copper rod and roller can be characterized by the binding energy between micro atoms. This mechanism is consistent with the strong plastic flow of metal in the groove of the rolling mill, which promotes stronger adhesion to the surface of the groove [30,31].

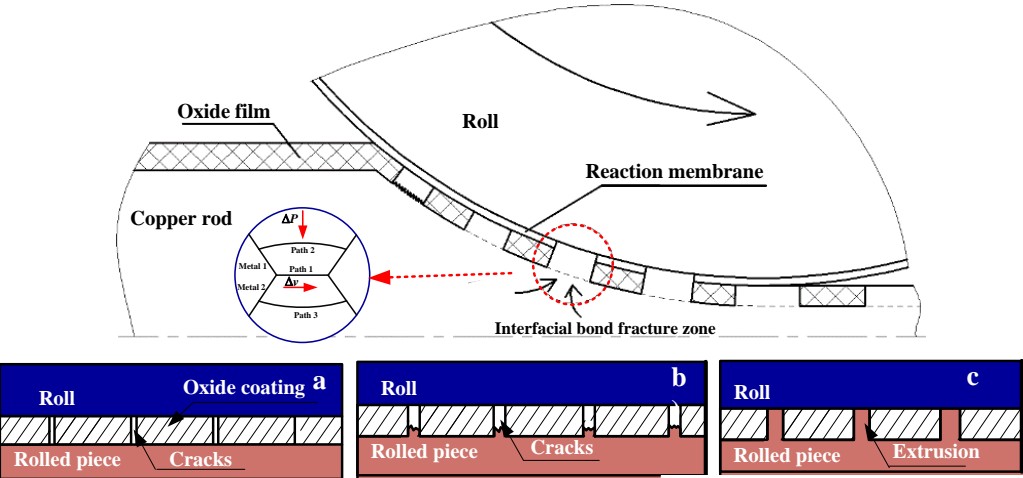

**Figure 5.** Schematic diagram of the mechanism of the roll sticking to the rolling copper rod.

The main component of the copper rod is the Cu atom and, the internal binding energy is (Cu-Cu); the main component of the roll is the Fe atom and, the internal binding energy is (Fe-Fe), and the interface energy between the copper rod and the roll surface is (Fe-Cu). Research shows that the interface energy of Fe-Cu is greater than that of metal, and the internal binding energy of Cu-Cu is the minimum [32]. Therefore, the fracture point of welding occurs on the copper matrix with the minimum internal binding energy, which is characterized by the copper sticking phenomenon. At the same time, the temperature, friction stress, and normal pressure of the surface layer and sliding layer of the contact surface between the roll and the rolled piece during the hot rolling process—in addition to the degree of deformation—have an important influence on the adhesion behavior of copper [33].

### 3.2. Rolling Adhesion Model

The rolling force is expressed by the product of the average unit pressure and the area of the deformation zone. The average unit pressure of pass rolling is characterized by the stress state coefficient and deformation resistance. The shape, size, and rolling calculation in the deformation zone of the 2# stand pass roll of the SCR3000 continuous casting and rolling line can be expressed by independent parameters. The friction model based on equivalent stress is adopted due to the large plastic deformation and the large positive pressure of the contact interface between the roll and the workpiece [34]:

$$\sigma_{fr} \leq -\mu \cdot \frac{\overline{\sigma}}{\sqrt{3}} \cdot \frac{2}{\pi} \arctan\left(\frac{v_r}{v_c}\right) \cdot \frac{v_r}{|v_c|} \tag{1}$$

where $\sigma_{fr}$ is the tangential friction stress, MPa; $\mu$ is the friction coefficient; $\overline{\sigma}$ is the equivalent stress, MPa; vr is the relative sliding speed of the two surfaces, mm/s; and $v_c$ is the critical relative speed between the contact bodies when sliding, 1 mm/s. The average deformation

degree of elliptical round pass rolling is related to the rolling process, and the average strain and strain rate are shown in Equation (2):

$$
\begin{cases}
\varepsilon = \frac{2}{3} \cdot \frac{H_0 - H_1}{H_0} \\
\dot{\varepsilon} = 0.105 n \sqrt{\frac{\varepsilon \cdot D}{2H_0}}
\end{cases}
\tag{2}
$$

where $n$ is roll speed, $H_0$ is the height before rolling; $H_1$ is the height after rolling; and D is the working diameter of the roll. The value for each parameter is presented in Tables 2 and 3.

**Table 3.** Matching table for extension coefficient of stands of continuous rolling mill in the SCR3000 production line.

| Production Speed | | 25.03 t/h | Billet Speed 0.216 m/s | | Bar Speed 16.349 m/s | | Casting Speed 1037 r/min | |
|---|---|---|---|---|---|---|---|---|
| Frame | D/mm | Rolling area(mm$^2$) | Revolution speed(r/min) | | Stackingratio | Stacking rate | Elongation coefficient | |
| | | | motor | roll | | | Theoretical | Actual |
| 1H | 304.80 | 2500 | 1585 | 22.11 | – | – | – | – |
| 2V | 304.80 | 1320 | 1598 | 44.06 | 1.0420 | 4.2% | 1.899 | 1.945 |
| 3H | 204.50 | 800 | 1603 | 101.99 | 0.9535 | −4.65% | 1.620 | 1.594 |
| 4V | 204.50 | 480 | 1662 | 169.19 | 0.9763 | −2.37% | 1.650 | 1.643 |
| 5H | 204.50 | 300 | 1619 | 257.56 | 0.9980 | −0.2% | 1.600 | 1.610 |
| 6V | 204.50 | 195 | 1624 | 401.19 | 0.9967 | −0.33% | 1.539 | 1.537 |
| 7H | 204.50 | 125 | 1727 | 611.11 | 1.0084 | 0.84% | 1.560 | 1.557 |
| 8V | 204.50 | 95 | 1590 | 819.17 | 1.0039 | 0.39% | 1.315 | 1.311 |
| 9H | 204.50 | 65 | 1637 | 1187.95 | 1.0111 | 1.11% | 1.462 | 1.417 |
| 10V | 204.50 | 50.27 | 1606 | 1574.51 | 1.0161 | 11.61% | 1.293 | 1.234 |

The average unit pressure P obtained by calculating the deformation resistance [35] of multi-element nonlinear copper in copper rod rolling is:

$$
P = 40.25 \dot{\varepsilon}^{0.0984} \varepsilon^{0.38111} e^{-0.001854T} \left( m + \frac{83.0}{m+8} - 8.90 \right) \left( 0.8 + \frac{0.8}{a_k} \right) (0.61 + 0.39\mu)
\tag{3}
$$

where $m$ is the shape coefficient of deformation area; ak is the pass axial ratio $a_k = R/(H cos\gamma)$, R = 21 mm, $\gamma = 30^0$ for the 2# roller; T is the rolling temperature; and $\mu$ is the friction index, which takes a value of 0.5~1.5 [36]

Due to the key practical problems of rolling, a reasonable simplification is made: the copper rod is isotropic and has uniform deformation, the real contact area between the copper rod and the impurity-free roller is equal to the welding area, and there is breakdown of the oxide coating.

When the rolling force is greater than or equal to the critical extrusion pressure, the rolled copper rod can completely extrude from the cracks of the oxide coating on the surface, which does not mean that the copper in the extruded part is in direct contact with the roll. The surface of the extruded copper rod matrix forms a reaction film with the emulsion to avoid direct contact between the copper rod and the roller. The reaction membrane has a certain bearing capacity beyond the bearing capacity, and the reaction membrane fails [37]. During rolling, the hot-rolled emulsion in the rolling zone forms a chemical reaction film with copper. The failure condition of the reaction film is reached when the amount of reaction film formation in the exposed zone is less than the amount of wear, so the critical state of the sticking roller is when the amount of reaction film formation is equal to the amount of wear of the reaction film. During rolling, the hot-rolled emulsion

in the rolling zone forms a chemical reaction film with copper. Therefore, the adhesion index model of copper sticking to roll is established, as shown in Equation (4):

$$\eta = \frac{S \cdot P}{S \cdot P_E + S' \cdot P_B} \tag{4}$$

where $\eta$ is the adhesion index; $S$ is the contact area of deformation area, (mm2); $S'$ is the extrusion area of a crack in deformation area (mm2); and $P_E$ and $P_B$ are critical extrusion stress of rolling force and critical stress of reaction membrane, respectively. The adhesion index $\eta$ indicates the difference between the rolling state and the critical state. When $\eta > 1$, the phenomenon of copper sticking on the roll will soon occur. When $\eta < 1$, the phenomenon of copper sticking to the roll will not occur. When $\eta = 1$, the critical state of copper sticking to the roll has been reached. When the adhesion index is greater than 1, the oxide coating and the reaction film will fail due to excessive rolling force, and the extruded copper in the crack of the coating will be affected by the effective normal contact pressure of the roll. According to the von Mises criterion, the real contact area of the contact surface in the process of sliding was analyzed, and the real contact area calculation model in the exposed area was obtained [38]. Considering the real contact area gain caused by the plastic deformation of the contact point under the interaction of the static and dynamic effective normal loads, the relationship between the real contact area of the exposed part in the deformation area and the effective normal contact pressure was modified [39]. According to the interaction between the models, the proportion model of the bond area of the copper rod was established, as shown in Equation (5):

$$A = f(D, H_1, H_0, \nu, \gamma, \xi, T, n, R, \mu, E, c_0) \tag{5}$$

where $\xi$ is the average oil film thickness; $R$ is the gas constant; $c_0$ is the concentration of the effective components in the emulsion.

### 3.3. Deformation Simulation Analysis of Copper Rod Rolled by Hot Continuous Rolling

Based on the analysis of on-site process parameters, a rolling model of SCR3000 hot strip rolling was established, and a numerical simulation of the stress–strain field, temperature field, and damage field during the hot rolling process was realized using the finite element method. The specific stress-strain field results are presented in Figure 6. When metal is plastically deformed, not only does the external shape change, but the internal structure and properties are also altered.

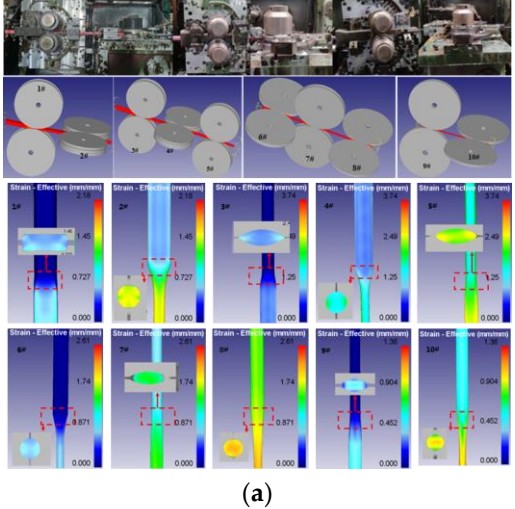

(**a**)

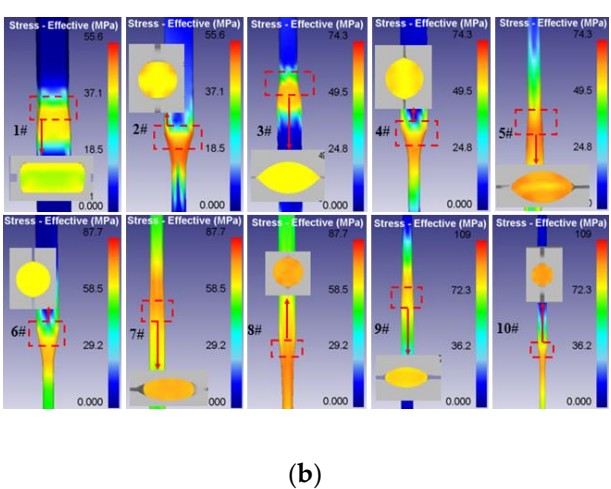

(**b**)

**Figure 6.** Numerical simulation of hot strip rolling in the SCR3000 line:(**a**) strain field; and (**b**) stress field.

The equivalent stress of the copper rod and its rolling interface during hot rolling were also simulated and analyzed. The equivalent stress program is presented in Figures 6b and 7. The deformation of roughly rolled copper rods mainly concentrated on the surface and the equivalent strain at the edge and corner is relatively large, which can easily lead to processing damage. The equivalent stress of the roll gap in the elliptical pass system for finishing copper rod is larger than that of the roll contact in the circular pass system. As rolling proceeds, the equivalent strain shifts from the surface to the center of the copper rod.

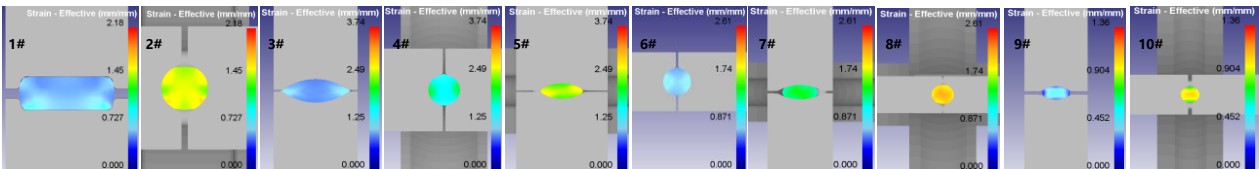

**Figure 7.** Cross-sectional strain evolution cloud diagram of each rolling pass.

During the entire rolling process, due to the difference in the amount of plastic deformation between the core metal and the surface metal of the rolled piece, and the difference in heat dissipation between the core metal and the surface metal, the surface metal is in contact with the roll, and convectively exchanges heat with the air, resulting in obvious heat dissipation. Therefore, the temperature of the cross section of the rolled piece is very different from the surface temperature (see Figure 8). The uniformity of the microstructure of the rolling section exerts a great influence on the overall performance of the final product, and the section uniformity after dynamic recrystallization is closely related to the uneven temperature distribution of the rolling section. Therefore, it is very important to analyze the temperature field distribution of the cross-section of the rolled product.

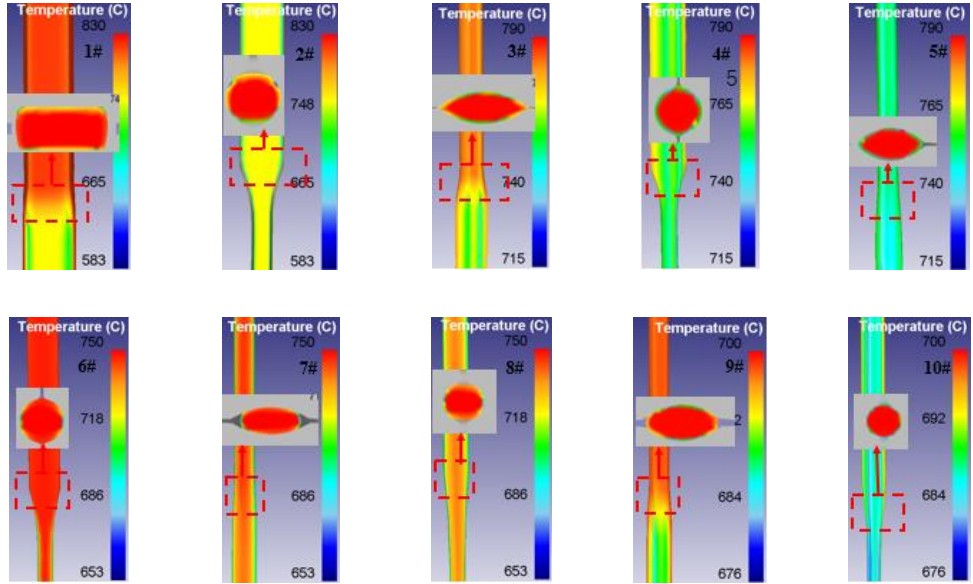

**Figure 8.** Equivalent temperature field of copper rod rolling and rolling sections in hot continuous rolling.

As rolling proceeds, the work hardening effect of the copper rod decreases with temperature, and the stress field increases significantly. The ductile damage to copper rods under large deformation, high temperature, and high strain rate rolling is an important problem in the plastic forming process of copper rods. By simulating the damage to copper rods and their cross-section in the continuous rolling process, the damage situation and damage distribution of copper rod in the continuous rolling process are obtained, which

can accurately predict the location of damage to a copper rod undergoing the hot rolling process, and can thus help to improve the quality of copper rod production (Figure 9).

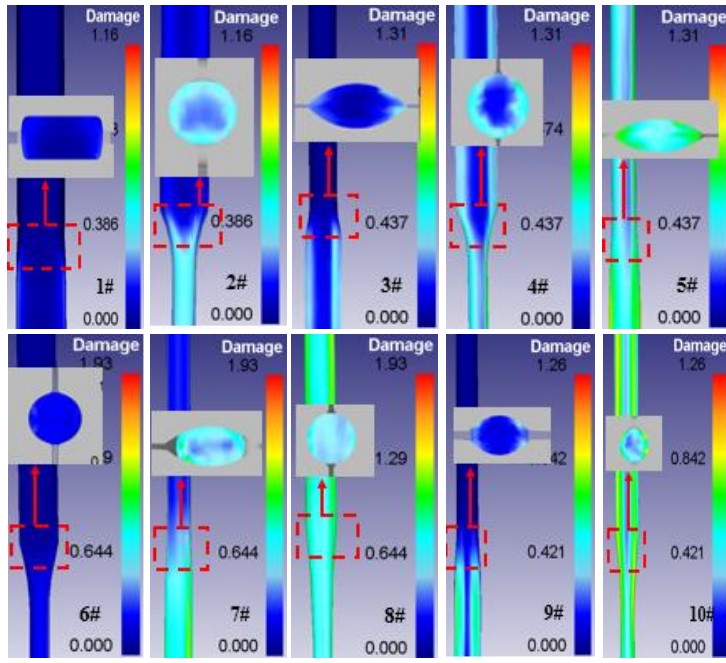

**Figure 9.** Damage program of hot continuous rolling copper rod and rolling section.

Based on investigation of the thermal deformation behavior of the Cu–Ag alloy, a finite element model of the continuous rolling process of SCR3000CuAg alloy wire rod was established with the help of large-scale finite element simulation of soft 3D deformation, and the continuous rolling process was numerically simulated in combination with the rolling process parameters. After calculation, the rolling process was studied by considering the stress and strain field evolution, temperature field change and damage field evolution, and the problems of the on-site continuous rolling process were demonstrated, revealing the basic law of hot continuous rolling deformation, and rolling of the Cu–Ag alloy wire. The evolution of performance combined with the copper sticking mechanism of the rolls in the hot continuous rolling process, the stress and strain fields, the temperature field, and the damage field were integrated in each pass during the severe deformation of hot rolling. The numerical simulation of the rolling deformation mechanism of each pass provided a solid foundation for solving the problem of copper sticking to the roll, and also provided theoretical support for the optimization of the production process of high-quality copper wire.

*3.4. Deformation Microstructure Evolution of Hot Continuous Rolling Copper Rod*

The metallographic microstructure observation of various passes can provide a clearer understanding of the microstructure evolution of the SCR3000 production line during the hot rolling process, and provide a theoretical basis for the subsequent improvement of this process, to facilitate better organization and control of the casting and rolling processes. The metallographic microstructure observation results of the copper bar cross-section of each pass are shown in Figure 10. After multi-pass rolling, the grain size of the copper billet decreases, and tends to homogenize. The grain size of the finished copper rod is about 20 μm. The microstructure of the copper alloys has an important influence on the performance, and determines the effect of rolling process optimization. It is an important research hotspot for copper alloys in the future. In order to better measure the rolling deformation law of copper wire rod and accurately predict the evolution of rolling performance, the hardness change of the copper alloy grain refining effect under the rolling

pass is considered. This will have a very important guiding role and practical significance for the optimization of copper alloy hot tandem rolling process.

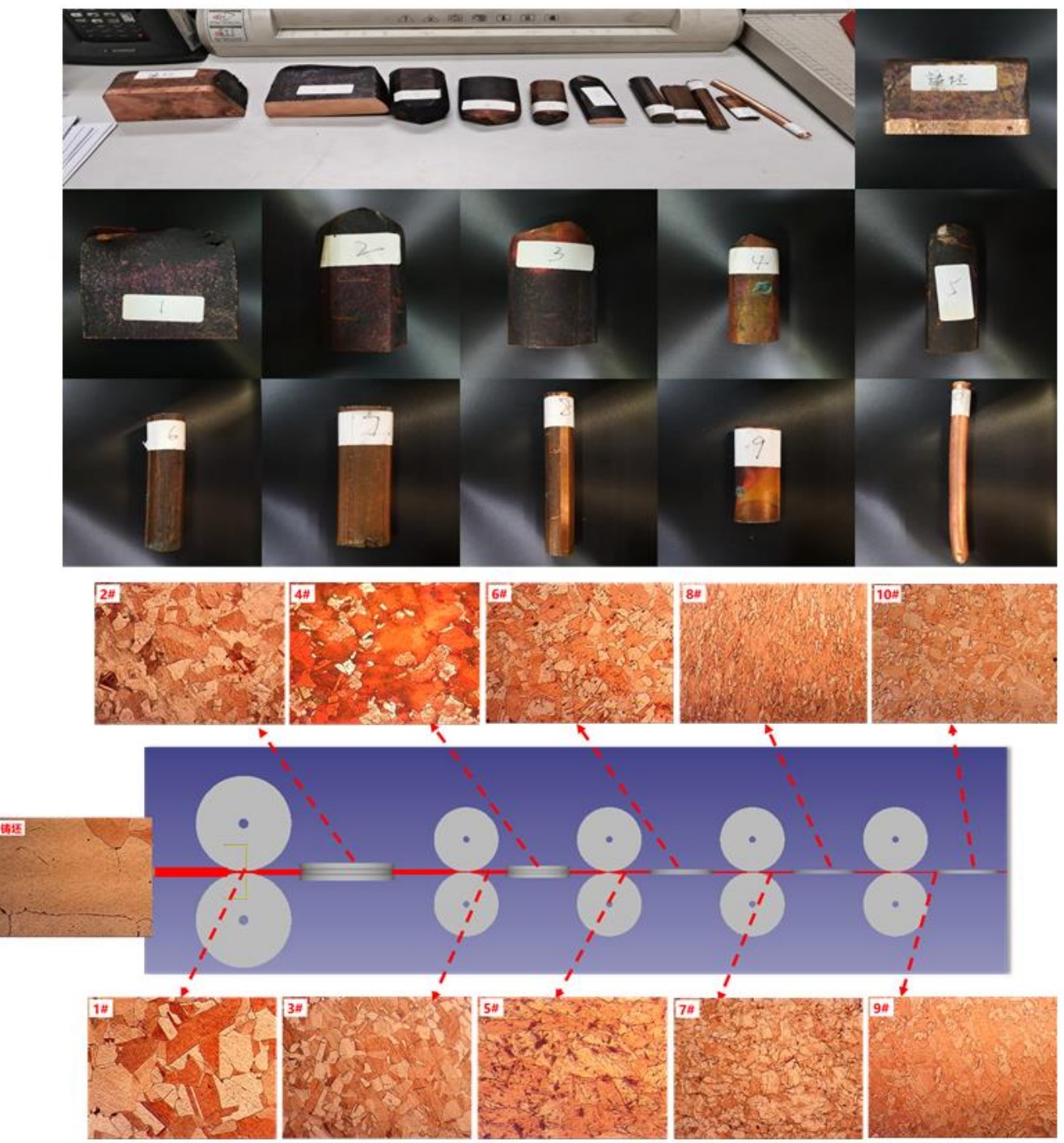

**Figure 10.** Microstructure and morphology of rolled copper rods in different passes.

## 4. Results and Discussion

Using the field process parameters (Table 3), combined with the rolling force model, a rolling simulation analysis of the 2# stand roll was carried out, and the rolling status is presented in Figure 11. Subsequent analysis revealed that 4.2% of the stacking ratio between the 1# stand and the 2# stand was in a state of strong tensile rolling, and −4.65% of the stacking ratio between the 2# stand and the 3# stand was in a state of strong stacking rolling.

The difference between the stacking ratio of the 2# stand is 8.85%, which is extremely poor. The finite element simulation analysis of hot continuous rolling demonstrated that the strain rate of the copper rod near the two ends of pass increases, the temperature decreases, and the surface damage increases during the rolling process of the 2# rolling mill; therefore, phenomenon of roller sticking is easily produced. The cause of roller sticking is explained by the rolling state.

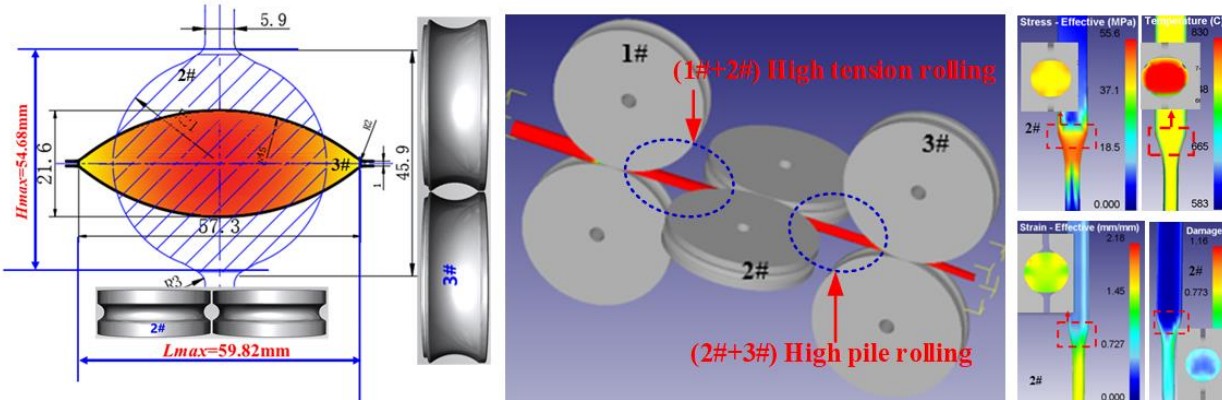

**Figure 11.** Simulation of rolling state of the 2# stand.

Based on the field data and the established roller sticking model, a rolling interface finite element model was developed. Based on the established finite element model, the normal stress field, temperature field, and bare rate of the copper rod surface in the deformation area were simulated and analyzed. The measurement value and process parameter value of the 2# roll pass were brought into the rolling roller sticking area ratio model, and the calculated value was compared with the field roller sticking area ratio of the 2# stand roll, as shown in Figure 12. The results demonstrate that the maximum contact normal stress is located at the junction of the two outer surfaces of the copper bar at the entrance of the deformation zone, and decreases gradually along the boundary of the contact interface. The maximum surface temperature of the copper bar occurs located at the junction of the two outer surfaces at the entrance of the deformation zone, and the temperature distribution decreases along the rolling direction [39]. The position of the maximum contact normal stress and maximum temperature correspond to those of the roller.

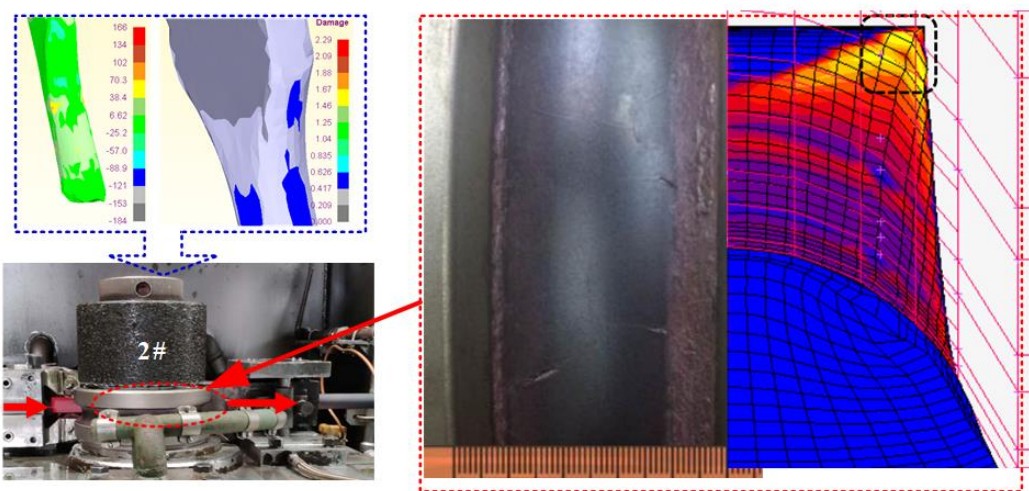

**Figure 12.** Simulation analysis of rolling interface state.

By investigating the influence of rolling process parameters, it was shown that the roll gap, temperature, roll speed, and lubricant concentration have an important influence on copper sticking to the roll, and that temperature has the greatest influence. Existing studies [39] have effectively solved the problem of copper sticking to rolls controlling the rolling temperature or via technical transformation of the emulsion spray device of the rolling mill. However, the effect was not stable, and copper sticking appeared 24 h later. Based on our research regarding the copper sticking behavior of rollers, we propose that the problem of copper sticking can be improved by adjusting the roller speeds and gap size employed in this process.

A prior study [40] simulated the deformation law of copper during the rolling process by establishing a hydrodynamic roll gap model of asymmetric friction conditions during copper hot rolling and its analytical solution, and clarified that the roll gap has an important effect on the plastic deformation of hot rolled copper. The poor rolling condition of the 2# stand copper rod in and out rolling was solved by considering the mechanism of copper sticking. Matching the motor speed to the elongation coefficient was proposed, and the motor speed of 1585 r/min and 1598 r/min were adjusted to 1549 r/min and 1586 r/min, respectively. The copper powder content of the copper rod decreased from 4.5 mg/250 mm to 3.9 mg/250 mm, which effectively improved the quality of the finished copper rod and optimized the copper sticking phenomenon to a certain extent. Based on the technical transformation of the emulsion spray device of the rolling mill, the main factor which influences copper sticking to the roller, i.e., the roll gap, was adjusted. As the copper bar of the 2# rolling mill is in a strong pull state and the copper bar is in a strong stacking force state—which seriously worsens the rolling section and aggravates the phenomenon of copper sticking—the roll gap was increased from the initial 5.75 mm to 5.90 mm, effectively solving the problem of copper sticking to the roll. This effect was stable, as shown in Figure 13. Meanwhile, the average copper powder content of the finished copper rod decreased from 4.3425 mg/250 mm to 3.3902 mg/250 mm, which greatly improved the quality of the rolled copper rod.

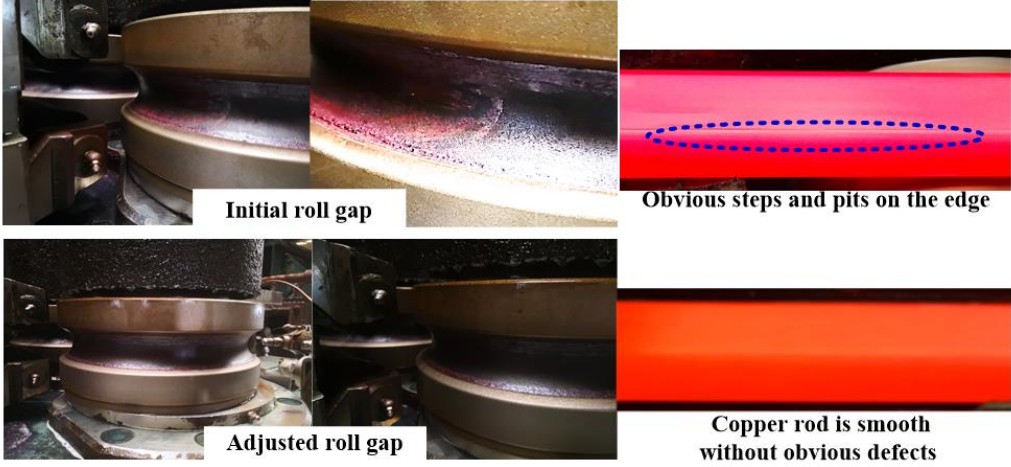

**Figure 13.** Regulating the 2# roll gap roll surface and copper rod surface.

## 5. Conclusions

(1) In this paper, the problem of copper sticking to the hot rolling roll of CuAg alloy wire rod in the SCR continuous casting and rolling production line was studied. Based on the principle of tribology, the binding energy of Cu-Cu is lower than the binding energy of Fe-Fe and Fe-Cu, and the Cu matrix is more likely to become a fracture site, which is the fundamental reason for the sticking to copper on the roll.

(2) The adhesion index model and adhesion area ratio model were established. Through numerical simulation analysis the rolling process, the temperature field and the

contact stress field of the hot rolling deformation zone were obtained, and were found to be consistent with the actual copper adhesion of the roll, which realized the characterization of the quantitative behavior of the roll adhesion.

(3) Combining the numerical analysis of the stress and strain fields, the temperature field and damage field during the deformation process of hot continuous rolling, the 2# rolling mill was found to be in a state of strong drawing and rolling, causing serious adhesion between the copper and the roll. A new method to match the motor speed with the elongation coefficient was proposed. The motor speed of the 1# and 2# rolling mills were adjusted to 1549 r/min and 1586 r/min, and the copper powder content was reduced from 4.3–4.8 mg/25 mm to 3.9 mg/25 mm, effectively reducing the copper powder content, and improving the quality of the Cu–Ag alloy wire.

(4) We proposed the following optimization measures for hot continuous rolling of Cu–Ag alloy wire: adjust the 2# roll gap from 5.75 mm to 5.90 mm, optimize the spray device, and reduce the copper powder content by 0.9523 mg/25 mm. These measures effectively and stably solved the problem of copper sticking.

(5) Effective methods and measures are provided to solve the problem of sticking rollers in the production process for wire rods made of copper or its alloys. These methods are expected to assist in the production of high-quality copper alloy wire rods. The technical "barriers" to solving the problem of copper sticking in the production process of high-quality copper alloy wires are addressed.

**Author Contributions:** Conceptualization, Y.L. and Y.P.; methodology, Y.L. and Y.P.; simulations, Y.L. and X.Q.; data curation, Y.L. and Y.P.; writing—original draft preparation, Y.L., Y.P. and X.Q.; writing—review and editing, Y.L. and Y.P.; project administration, Y.P.; All authors have read and agreed to the published version of the manuscript.

**Funding:** This research received no external funding.

**Institutional Review Board Statement:** Not applicable.

**Informed Consent Statement:** Not applicable.

**Data Availability Statement:** All relevant data are obtained from the article.

**Acknowledgments:** This work is supported by the Research Program supported by the National Basic Research Program of China(2017YFB0306402) and Key Projects of the Natural Fund of Hebei province (E2017203161).

**Conflicts of Interest:** The authors declare no conflict of interest.

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
