# Peer review of "Mechanism of and Key Technologies for Copper Bonding in the Hot Rolling of SCR Continuous Casting and Rolling"

_applsci, doi:10.3390/app112211023_

Round 1

Reviewer 1 Report

Reviewer: Comments and Suggestions for Authors

In this paper, a research on the mechanism of copper bonding in hot rolling of SCR continuous casting and rolling has been carried out. The research work should be published because the results are commercially and technologically important. However, a considerable amount of re-writing is required.

  1. There are some typing errors. English needs improvement.
  2. Literature review on the related subject is not complete; there are recent strong papers that are not discussed and cited.
  3. The materials and methods should be described in more detail.
  4. The authors did not sufficiently emphasize the scientific/technological contribution in the field or future work in the conclusion. More analysis and discussion of results is needed.
  5. The conclusion only summarizes the presented research.
  6. The information presented in the figure legends is very brief.
  7. Check units format.

Reviewer 2 Report

Good paper with potential industrial impact. Major revisions are in order for the authors to address the comments below:

Language needs to be improved several minor errors found.

“In recent years, with the rapid development of the national econ27 omy and industrial technology, higher requirements are put for28 ward for the demand and quality of copper poles for electrical en29 gineering”: not only copper but its alloys. See for example 10.1016/j.matdes.2016.03.032, 10.1016/j.matdes.2018.03.066 and 10.1016/j.scriptamat.2019.12.012 and further complement the introduction.

“The problem of sticking the roller on the surface of the roll exists in the rolling of ferrous and non-ferrous metals. A”: add references

“ent surface characteristics on ox”: what were these characteristics? Detail.

Its hard to read the paper because of the layout in two columns. Several lines are over the text!

“ved to solve the failure problem of the 10# roller”: why is this roller important? Clarify.

Fig 1 needs a scale bar. Same for fig 2.

“There are pits on the edge of the copper rod rolled by the sticking rolle”: cannot be seen. Add close up images?

Also what drives the pitting?

For the modelling where did the authors got the values for the constants? Unclear. Please add such information.

“s mainly concentrates on the surface and the equivalent strain at the edge and corner is relatively large, which”: was this expected? What is the significance. Needs to be discussed.

“ent strain shifts from the surface to the center of the cop”: it would be interesting to quantify these shifths.

“In the electron microscope observation of SEM, Cu2O was dispersed in the matrix, presumably due to in352 adequate oxygen content control”: the authors do not show any SEM images but only OM images. Clarify. Also, please increase the size of the photos to better see the grain size. Where is the evidence of Cu2O?

“parameters(table11)”: table 11??

Have the authors quantified the hardness changes and correlate it with the grain size? This would be useful.

Round 2

Reviewer 1 Report

The manuscript has been sufficiently improved to warrant publication in Applied Sciences

Reviewer 2 Report

The authors did a good job in revising the manuscript and acceptance is recommended.